# Is Attention Better Than Matrix Decomposition?

**Zhengyang Geng**[1,2], **Meng-Hao Guo**[3,*] **Hongxu Chen**[4], **Xia Li**[2], **Ke Wei**[4], **Zhouchen Lin**[2,5] [†]

[1]Zhejiang Lab; [2]Key Lab. of Machine Perception (MoE), School of EECS, Peking University;
[3]Tsinghua University; [4]School of Data Science, Fudan University; [5]Pazhou Lab

## Abstract

As an essential ingredient of modern deep learning, attention mechanism, especially self-attention, plays a vital role in the global correlation discovery. However, is hand-crafted attention irreplaceable when modeling the global context? Our intriguing finding is that self-attention is not better than the matrix decomposition (MD) model developed 20 years ago regarding the performance and computational cost for encoding the long-distance dependencies. We model the global context issue as a low-rank completion problem and show that its optimization algorithms can help design global information blocks. This paper then proposes a series of Hamburgers, in which we employ the optimization algorithms for solving MDs to factorize the input representations into sub-matrices and reconstruct a low-rank embedding. Hamburgers with different MDs can perform favorably against the popular global context module self-attention when carefully coping with gradients back-propagated through MDs. Comprehensive experiments are conducted in the vision tasks where it is crucial to learn the global context, including semantic segmentation and image generation, demonstrating significant improvements over self-attention and its variants. Code is available.

## 1 Introduction

Since self-attention and transformer (Vaswani et al., 2017) showed significant advantages over recurrent neural networks and convolutional neural networks in capturing long-distance dependencies, attention has been widely adopted by computer vision (Wang et al., 2018; Zhang et al., 2019a) and natural language processing (Devlin et al., 2019) for global information mining. However, is hand-crafted attention irreplaceable when modeling the global context?

This paper focuses on a new approach to design global context modules. The key idea is, *if we formulate the inductive bias like the global context into an objective function, the optimization algorithm to minimize the objective function can construct a computational graph, i.e., the architecture we need in the networks*. We particularize this idea by developing a counterpart for the most representative global context module, self-attention. Considering extracting global information in the networks as finding a dictionary and the corresponding codes to capture the inherent correlation, we model the context discovery as low-rank completion of the input tensor and solve it via matrix decomposition. This paper then proposes a global correlation block, Hamburger, by employing matrix decomposition to factorize the learned representation into sub-matrices so as to recover the clean low-rank signal subspace. The iterative optimization algorithm to solve matrix decomposition defines the central computational graph, *i.e.,* Hamburger's architecture.

Our work takes advantage of the matrix decomposition models as the foundation of Hamburger, including Vector Quantization (VQ) (Gray & Neuhoff, 1998), Concept Decomposition (CD) (Dhillon & Modha, 2001), and Non-negative Matrix Factorization (NMF) (Lee & Seung, 1999). Additionally, instead of directly applying Back-Propagation Through Time (BPTT) algorithm (Werbos et al., 1990) to differentiate the iterative optimization, we adopt a truncated BPTT algorithm, *i.e.,* one-step gradient, to back-propagate the gradient effectively. We illustrate the advantages of Hamburger in

---

[*]Equal first authorship
[†]Corresponding author

the fundamental vision tasks where global information has been proven crucial, including semantic segmentation and image generation. The experiments prove that optimization-designed Hamburger can perform competitively with state-of-the-art attention models when avoiding the unstable gradient back-propagated through the iterative computational graph of MD. Hamburger sets new state-of-the-art records on the PASCAL VOC dataset (Everingham et al., 2010) and PASCAL Context dataset (Mottaghi et al., 2014) for semantic segmentation and surpasses existing attention modules for GANs in the large scale image generation on ImageNet (Deng et al., 2009).

The contributions of this paper are listed as follows:

- We show a white-box approach to design global information blocks, *i.e.,* by turning the optimization algorithm that minimizes an objective function, in which modeling the global correlation is formulated as a low-rank completion problem, into the architecture.
- We propose Hamburger, a light yet powerful global context module with $\mathcal{O}(n)$ complexity, surpassing various attention modules on semantic segmentation and image generation.
- We figure out that the main obstacle of applying MD in the networks is the unstable backward gradient through its iterative optimization algorithm. As a pragmatic solution, the proposed one-step gradient facilitates the training of Hamburger with MDs.

## 2 METHODOLOGY

### 2.1 WARM UP

Since matrix decomposition is pivotal to the proposed Hamburger, we first review the idea of matrix decomposition. A common view is that matrix decomposition factorizes the observed matrix into a product of several sub-matrices, *e.g.,* Singular Value Decomposition. However, a more illuminating perspective is that, by assuming the generation process, matrix decomposition acts as the inverse of the generation, disassembling the atoms that make up the complex data. From the reconstruction of the original matrices, matrix decomposition recovers the latent structure of observed data.

Suppose that the given data are arranged as the columns of a large matrix $\boldsymbol{X} = [\mathbf{x}_1, \cdots, \mathbf{x}_n] \in \mathbb{R}^{d \times n}$. A general assumption is that there is a low-dimensional subspace, or a union of multiple subspaces hidden in $\boldsymbol{X}$. That is, there exists a dictionary matrix $\boldsymbol{D} = [\mathbf{d}_1, \cdots, \mathbf{d}_r] \in \mathbb{R}^{d \times r}$ and corresponding codes $\boldsymbol{C} = [\mathbf{c}_1, \cdots, \mathbf{c}_n] \in \mathbb{R}^{r \times n}$ that $\boldsymbol{X}$ can be expressed as

$$\overset{\overleftarrow{generation}}{\boldsymbol{X} = \bar{\boldsymbol{X}} + \boldsymbol{E} = \boldsymbol{D}\boldsymbol{C} + \boldsymbol{E},}_{\underrightarrow{decomposition}} \tag{1}$$

where $\bar{\boldsymbol{X}} \in \mathbb{R}^{d \times n}$ is the output low-rank reconstruction, and $\boldsymbol{E} \in \mathbb{R}^{d \times n}$ is the noise matrix to be discarded. Here we assume that the recovered matrix $\bar{\boldsymbol{X}}$ has the low-rank property, such that

$$\text{rank}(\bar{\boldsymbol{X}}) \leq \min(\text{rank}(\boldsymbol{D}), \text{rank}(\boldsymbol{C})) \leq r \ll \min(d, n). \tag{2}$$

Different MDs can be derived by assuming structures to matrices $\boldsymbol{D}$, $\boldsymbol{C}$, and $\boldsymbol{E}$ (Kolda & Bader, 2009; Udell et al., 2016). MD is usually formulated as an objective with various constraints and then solved by optimization algorithms, with classic applications to image denoising (Wright et al., 2009; Lu et al., 2014), inpainting (Mairal et al., 2010), and feature extraction (Zhang et al., 2012).

### 2.2 PROPOSED METHOD

We focus on building global context modules for the networks without painstaking hand-crafted design. Before starting our discussion, we review the representative hand-designed context block self-attention pithily.

The attention mechanism aims at finding a group of concepts for further conscious reasoning from massive unconscious context (Xu et al., 2015; Bengio, 2017; Goyal et al., 2019). As a representative, self-attention (Vaswani et al., 2017) is proposed for learning long-range dependencies in machine translation,

$$\text{Attention}\,(\boldsymbol{Q}, \boldsymbol{K}, \boldsymbol{V}) = \text{softmax}\left(\frac{\boldsymbol{Q}\boldsymbol{K}^\top}{\sqrt{d}}\right)\boldsymbol{V}, \tag{3}$$

where $\boldsymbol{Q}, \boldsymbol{K}, \boldsymbol{V} \in \mathbb{R}^{n \times d}$ are features projected by linear transformations from the input. Self-attention extracts global information via attending all tokens at a time rather than the typical one-by-one processing of recurrent neural networks.

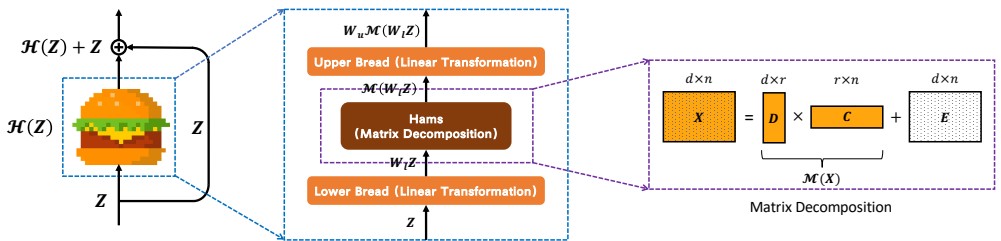

Figure 1: Overview of Hamburger

Though self-attention and its variants achieved great success, researchers are confronted with (1) developing new global context modules based on self-attention, typically via hand-crafted engineering, and (2) explaining why current attention models work. This paper bypasses both issues and finds a method to easily design global context modules via a well-defined white-box toolkit. We try to formulate the human inductive bias, like the global context, as an objective function and use the optimization algorithm to solve such a problem to design the module's architecture. The optimization algorithm creates a computational graph, takes some input, and finally outputs the solution. We apply the computational graph of optimization algorithms for the central part of our context module.

Based on this approach, we need to model the networks' global information issue as an optimization problem. Take the convolutional neural networks (CNN) as an example for further discussion. The networks output a tensor $\mathcal{X} \in \mathbb{R}^{C \times H \times W}$ after we feed into an image. Since the tensor can be seen as a set of $HW$ $C$-dimensional hyper-pixels, we unfold the tensor into a matrix $\boldsymbol{X} \in \mathbb{R}^{C \times HW}$. When the module learns the long-range dependencies or the global context, the hidden assumption is that the hyper-pixels are inherently correlated. For the sake of simplicity, we assume that hyper-pixels are linearly dependent, which means that each hyper-pixel in $\boldsymbol{X}$ can be expressed as the linear combination of bases whose elements are typically much less than $HW$. In the ideal situation, the global information hidden in $\boldsymbol{X}$ can be *low-rank*. However, due to vanilla CNN's poor ability to model the global context (Wang et al., 2018; Zhang et al., 2019a), the learned $\boldsymbol{X}$ is usually corrupted with redundant information or incompleteness. The above analysis suggests a potential method to model the global context, *i.e.,* by completing the low-rank part $\bar{\boldsymbol{X}}$ in the unfolded matrix $\boldsymbol{X}$ and discarding the noise part $\boldsymbol{E}$, using the classic matrix decomposition models described in Eq. (1), which filters out the redundancy and incompleteness at the same time. We thus model learning the global context as a low-rank completion problem with matrix decomposition as its solution. Using the notion of Sec. 2.1, the general objective function of matrix decomposition is

$$\min_{\boldsymbol{D}, \boldsymbol{C}} \mathcal{L}(\boldsymbol{X}, \boldsymbol{DC}) + \mathcal{R}_1(\boldsymbol{D}) + \mathcal{R}_2(\boldsymbol{C}) \tag{4}$$

where $\mathcal{L}$ is the reconstruction loss, $\mathcal{R}_1$ and $\mathcal{R}_2$ are regularization terms for the dictionary $\boldsymbol{D}$ and the codes $\boldsymbol{C}$. Denote the optimization algorithm to minimize Eq. (4) as $\mathcal{M}$. $\mathcal{M}$ is the core architecture we deploy in our global context module. To help readers further understand this modeling, We also provide a more intuitive illustration in Appendix G.

In the later sections, we introduce our global context block, Hamburger, and then discuss detailed MD models and optimization algorithms for $\mathcal{M}$. Finally, we handle the gradient issue for back-propagation through matrix decomposition.

### 2.2.1 HAMBURGER

Hamburger consists of one slice of "ham" (matrix decomposition) and two slices of "bread" (linear transformation). As the name implies, Hamburger first maps the input $\boldsymbol{Z} \in \mathbb{R}^{d_z \times n}$ into feature space with a linear transformation $\boldsymbol{W}_l \in \mathbb{R}^{d \times d_z}$, namely "lower bread", then uses matrix decomposition $\mathcal{M}$ to solve a low-rank signal subspace, corresponding to the "ham", and finally transforms extracted signals into the output with another linear transformation $\boldsymbol{W}_u \in \mathbb{R}^{d_z \times d}$, called "upper bread",

$$\mathcal{H}(\boldsymbol{Z}) = \boldsymbol{W}_u \mathcal{M}(\boldsymbol{W}_l \boldsymbol{Z}), \tag{5}$$

where $\mathcal{M}$ is matrix decomposition to recover the clear latent structure, functioning as a *global nonlinearity*. Detailed architectures of $\mathcal{M}$, *i.e.,* optimization algorithms to factorize $\boldsymbol{X}$, are discussed in Sec. 2.2.2. Fig. 1 describes the architecture of Hamburger, where it collaborates with the networks via Batch Normalization (BN) (Ioffe & Szegedy, 2015), a skip connection, and finally outputs $\boldsymbol{Y}$,

$$\boldsymbol{Y} = \boldsymbol{Z} + \text{BN}(\mathcal{H}(\boldsymbol{Z})). \tag{6}$$

### 2.2.2 HAMS

This section describes the structure of "ham", *i.e.,* $\mathcal{M}$ in Eq. (5). As discussed in the previous section, by formulating the global information discovery as an optimization problem of MD, algorithms to solve MD naturally compose $\mathcal{M}$. $\mathcal{M}$ takes the output of "lower bread" as its input and computes a low-rank reconstruction as its output, denoted as $\boldsymbol{X}$ and $\bar{\boldsymbol{X}}$, respectively.

$$\mathcal{M}(\boldsymbol{X}) = \bar{\boldsymbol{X}} = \boldsymbol{DC}. \tag{7}$$

We investigate two MD models for $\mathcal{M}$, Vector Quantization (VQ), and Non-negative Matrix Factorization (NMF) to solve $\boldsymbol{D}$ and $\boldsymbol{C}$ and reconstruct $\bar{\boldsymbol{X}}$, while leaving Concept Decomposition (CD) to Appendix B. The selected MD models are introduced briefly because we endeavor to illustrate the importance of the low-rank inductive bias and the optimization-driven designing method for global context modules rather than any specific MD models. It is preferred to abstract the MD part as a whole, *i.e.,* $\mathcal{M}$ in the context of this paper, and focus on how Hamburger can show the superiority in its entirety.

**Vector Quantization** Vector Quantization (VQ) (Gray & Neuhoff, 1998), a classic data compression algorithm, can be formulated as an optimization problem in term of matrix decomposition:

$$\min_{\boldsymbol{D,C}} \|\boldsymbol{X} - \boldsymbol{DC}\|_F \quad \text{s.t. } \mathbf{c}_i \in \{\mathbf{e}_1, \mathbf{e}_2, \cdots, \mathbf{e}_r\}, \tag{8}$$

where $\boldsymbol{e}_i$ is the canonical basis vector, $\mathbf{e}_i = [0, \cdots, \underset{i\text{th}}{1}, \cdots, 0]^\top$. The solution to minimize the objective in Eq. (8) is K-means (Gray & Neuhoff, 1998). However, to ensure that VQ is differentiable, we replace the hard $\arg\min$ and Euclidean distance with $softmax$ and $cosine$ similarity, leading to Alg. 1, where $cosine(\boldsymbol{D}, \boldsymbol{X})$ is a similarity matrix whose entries satisfy $cosine(\boldsymbol{D}, \boldsymbol{X})_{ij} = \frac{\mathbf{d}_i^\top \mathbf{x}_j}{\|\mathbf{d}\|\|\mathbf{x}\|}$, and $softmax$ is applied column-wise and $T$ is the temperature. Further we can obtain a hard assignment by a one-hot vector when $T \to 0$.

---

| **Algorithm 1** Ham: Soft VQ | **Algorithm 2** Ham: NMF with MU |
|---|---|
| Input $\boldsymbol{X}$. Initialize $\boldsymbol{D}, \boldsymbol{C}$. | Input $\boldsymbol{X}$. Initialize non-negative $\boldsymbol{D}, \boldsymbol{C}$ |
| **for** $k$ from 1 to $K$ **do** | **for** $k$ from 1 to $K$ **do** |
| $\quad \boldsymbol{C} \leftarrow softmax(\frac{1}{T}cosine(\boldsymbol{D}, \boldsymbol{X}))$ | $\quad \boldsymbol{C}_{ij} \leftarrow \boldsymbol{C}_{ij} \frac{(\boldsymbol{D}^\top \boldsymbol{X})_{ij}}{(\boldsymbol{D}^\top \boldsymbol{D} \boldsymbol{C})_{ij}}$ |
| $\quad \boldsymbol{D} \leftarrow \boldsymbol{X}\boldsymbol{C}^\top diag(\boldsymbol{C}\mathbf{1}_n)^{-1}$ | $\quad \boldsymbol{D}_{ij} \leftarrow \boldsymbol{D}_{ij} \frac{(\boldsymbol{X}\boldsymbol{C}^\top)_{ij}}{(\boldsymbol{D}\boldsymbol{C}\boldsymbol{C}^\top)_{ij}}$ |
| **end for** | **end for** |
| Output $\bar{\boldsymbol{X}} = \boldsymbol{DC}$. | Output $\bar{\boldsymbol{X}} = \boldsymbol{DC}$. |

---

**Non-negative Matrix Factorization** If we impose non-negative constraints on the dictionary $\boldsymbol{D}$ and the codes $\boldsymbol{C}$, it leads to Non-negative Matrix Factorization (NMF) (Lee & Seung, 1999):

$$\min_{\boldsymbol{D,C}} \|\boldsymbol{X} - \boldsymbol{DC}\|_F \quad \text{s.t. } \boldsymbol{D}_{ij} \geq 0, \boldsymbol{C}_{jk} \geq 0. \tag{9}$$

To satisfy the non-negative constraints, we add a ReLU non-linearity before putting $\boldsymbol{X}$ into NMF. We apply the Multiplicative Update (MU) rules (Lee & Seung, 2001) in Alg. 2 to solve NMF, which guarantees the convergence.

As white-box global context modules, VQ, CD, and NMF are straightforward and light, showing remarkable efficiency. They are formulated into optimization algorithms that mainly consist of matrix multiplications with the complexity $\mathcal{O}(ndr)$, much cheaper than complexity $\mathcal{O}(n^2d)$ in self-attention as $r \ll n$. All three MDs are memory-friendly since they avoid generating a large $n \times n$ matrix as an intermediate variable, like the product of $\boldsymbol{Q}$ and $\boldsymbol{K}$ of self-attention in Eq. (3). In the later section, our experiments prove MDs are at least on par with self-attention, though the architectures of $\mathcal{M}$ are created by optimization and look different from classic dot product self-attention.

### 2.3 ONE-STEP GRADIENT

Since $\mathcal{M}$ involves an optimization algorithm as its computational graph, a crux to fuse it into the networks is how the iterative algorithm back-propagates gradient. The RNN-like behavior of optimization suggests Back-Propagation Through Time (BPTT) algorithm (Werbos et al., 1990) as the standard choice to differentiate the iterative process. We first review the BPTT algorithm below. However, in practice, the unstable gradient from BPTT does harm Hamburger's performances. Hence we build an abstract model to analyze the drawbacks of BPTT and try to find a pragmatic solution while considering MD's nature as an optimization algorithm.

As shown in Fig. 2, $\mathbf{x}$, $\mathbf{y}$ and $\mathbf{h}^t$ denote input, output and intermediate result at time step $t$, respectively, while $\mathcal{F}$ and $\mathcal{G}$ are operators. At each time step, the model receives the same input $\mathbf{x}$ processed by the underlying networks.

$$\mathbf{h}^{t+1} = \mathcal{F}(\mathbf{h}^t, \mathbf{x}). \tag{10}$$

The intermediate results $\mathbf{h}^i$ are all discarded. Only the output of the last step $\mathbf{h}^t$ is passed through $\mathcal{G}$ for output $\mathbf{y}$,

$$\mathbf{y} = \mathcal{G}(\mathbf{h}^t). \tag{11}$$

In the BPPT algorithm, the gradient from output $\mathbf{y}$ to input $\mathbf{x}$ is given, according to the Chain rule:

$$\frac{\partial \mathbf{y}}{\partial \mathbf{x}} = \sum_{i=0}^{t-1} \frac{\partial \mathbf{y}}{\partial \mathbf{h}^t} \left( \prod_{j=t-i}^{t-1} \frac{\partial \mathbf{h}^{j+1}}{\partial \mathbf{h}^j} \right) \frac{\partial \mathbf{h}^{t-i}}{\partial \mathbf{x}}. \tag{12}$$

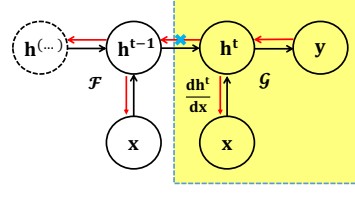

A thought experiment is to consider $t \to \infty$, leading to a fully converged result $\mathbf{h}^*$ and infinite terms in Eq. (12). We suppose

Figure 2: One-step Gradient

that both $\mathcal{F}$ and $\mathcal{G}$ are Lipschitz with constants $L_h$ w.r.t. $\mathbf{h}$, $L_x$ w.r.t. $\mathbf{x}$, and $L_{\mathcal{G}}$, and $L_h < 1$. Note that these assumptions apply to a large number of optimization or numerical methods. Then we have:

**Proposition 1** $\{\mathbf{h}^i\}_t$ has linear convergence.

**Proposition 2** $\lim_{t \to \infty} \frac{\partial \mathbf{y}}{\partial \mathbf{x}} = \frac{\partial \mathbf{y}}{\partial \mathbf{h}^*} (\boldsymbol{I} - \frac{\partial \mathcal{F}}{\partial \mathbf{h}^*})^{-1} \frac{\partial \mathcal{F}}{\partial \mathbf{x}}$.

**Proposition 3** $\lim_{t \to \infty} \|\frac{\partial \mathbf{y}}{\partial \mathbf{h}^0}\| = 0$, $\lim_{t \to \infty} \|\frac{\partial \mathbf{y}}{\partial \mathbf{x}}\| \leq \frac{L_{\mathcal{G}} L_x}{1-L_h}$.

Table 1: One-step Gradient & BPTT

| Method | One-step | BPTT |
|--------|----------|------|
| VQ | 77.7(77.4) | 76.6(76.3) |
| CD | 78.1(77.5) | 75.0(74.6) |
| NMF | 78.3(77.8) | 77.4(77.0) |

It is easy to incur gradient vanishing w.r.t. $\mathbf{h}^0$ when $L_h$ is close to 0 and gradient explosion w.r.t. $\mathbf{x}$ when $L_h$ is close to 1. The Jacobian matrix $\frac{\partial \mathbf{y}}{\partial \mathbf{x}}$, moreover, suffers from an ill-conditioned term $(\boldsymbol{I} - \frac{\partial \mathcal{F}}{\partial \mathbf{h}^*})^{-1}$ when the largest eigenvalue of $\frac{\partial \mathcal{F}}{\partial \mathbf{h}}$, i.e., the Lipschitz constant of $\mathcal{F}$ w.r.t. $\mathbf{h}$, approaches 1 and its minimal eigenvalue typically stays near 0, thus restricts the capability of the gradient to search the well-generalized solution in the parameter space. The erratic scale and spectrum of the gradient back through the optimization algorithm indicate the infeasibility to apply BPTT to Hamburger directly, corroborated by the experiments in Tab. 1, using the same ablation settings as Sec. 3.1.

The analysis inspires us a possible solution. Note that *there are a multiplication of multiple Jacobian matrices $\frac{\partial \mathbf{h}^j}{\partial \mathbf{h}^{j-1}}$ and a summation of an infinite series in BPTT algorithm*, leading to uncontrollable scales of gradients. It enlightens us to drop some minor terms in the gradient while preserving its dominant terms to ensure the direction is approximately right. Considering terms of Eq. (12) as a series, i.e., $\{\frac{\partial \mathbf{y}}{\partial \mathbf{h}^t} \left( \prod_{j=t-i}^{t-1} \frac{\partial \mathbf{h}^{j+1}}{\partial \mathbf{h}^j} \right) \frac{\partial \mathbf{h}^{t-i}}{\partial \mathbf{x}}\}_i$, it makes sense to use the first term of this series to approximate the gradient if the scale of its terms decays exponentially measured by the operator norm. The first term of the gradient is from the last step of optimization, leading to the one-step gradient,

$$\widehat{\frac{\partial \mathbf{y}}{\partial \mathbf{x}}} = \frac{\partial \mathbf{y}}{\partial \mathbf{h}^t} \frac{\partial \mathbf{h}^t}{\partial \mathbf{x}}. \tag{13}$$

The one-step gradient is a linear approximation of the BPTT algorithm when $t \to \infty$ according to the Proposition 2. It is easy to implement, requiring a $no\_grad$ operation in PyTorch (Paszke et al., 2019) or $stop\_gradient$ operation in TensorFlow (Abadi et al., 2016) and reducing the time and space complexity from $\mathcal{O}(t)$ in BPTT to $\mathcal{O}(1)$. We test adding more terms to the gradient but its performance is worse than using one step. According to experimental results, one-step gradient is acceptable to back-propagate gradient through MDs.

Table 2: Ablation on components of Hamburger with NMF Ham.

| Method | mIoU(%) | Params |
|---|---|---|
| baseline | 75.9(75.7) | 32.67M |
| basic | 78.3(77.8) | +0.50M |
| - ham | 75.8(75.6) | +0.50M |
| - upper bread | 77.0(76.8) | +0.25M |
| - lower bread | 77.3(77.2) | +0.25M |
| only ham | 77.0(76.8) | +0M |

## 3 EXPERIMENTS

In this section we present experimental results demonstrating the techniques described above. Two vision tasks that benefit a lot from global information and attention mechanism attract us, including semantic segmentation (over 50 papers using attention) and deep generative models like GANs (most state-of-the-art GANs adopt self-attention since SAGAN (Zhang et al., 2019a)). Both tasks are highly competitive and thus enough for comparing Hamburger with self-attention. Ablation studies show the importance of MD in Hamburger as well as the necessity of the one-step gradient. We emphasize the superiority of Hamburger on modeling global context over self-attention regarding both performance and computational cost.

### 3.1 ABLATION EXPERIMENTS

We choose to conduct all ablation experiments on the PASCAL VOC dataset (Everingham et al., 2010) for semantic segmentation, and report mIoU of *5 runs* on the validation set in the form of $\mathrm{best}(\mathrm{mean})$. ResNet-50 (He et al., 2016) with output stride 16 is the backbone for all ablation experiments. We employ a $3\times3$ conv with BN (Ioffe & Szegedy, 2015) and ReLU to reduce channels from 2048 to 512 and then add Hamburger, the same location as popular attentions in semantic segmentation. For detailed training settings, please see Appendix E.1.

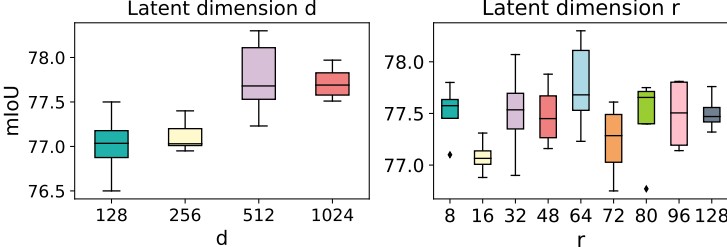

Figure 3: Ablation on $d$ and $r$

**Breads and Hams**  We ablate each part of the Hamburger. Removing MD (ham) causes the most severe decay in performance, attesting to the importance of MD. Even if only the parameter-free MD is added (only ham), the performance can visibly improve. Parameterization also helps the Hamburger process the extracted features. Bread, especially upper bread, contributes considerable performance.

**Latent Dimension $d$ and $r$**  It is worth noting that there is no simple linear relation between $d$ and $r$ with performances measured by mIoU, though $d = 8r$ is a satisfactory choice. Experiments show that even $r = 8$ performs well, revealing that it can be very cheap for modeling the global context.

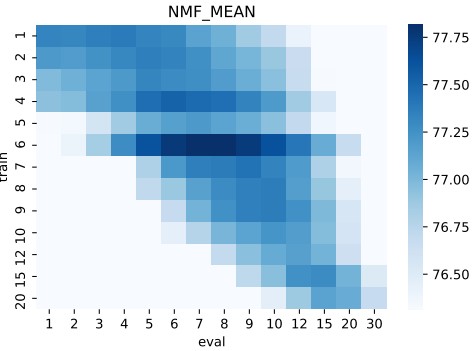

Figure 4: Ablation on $K$

**Iterations** $K$   We test more optimization steps in the evaluation stage. In general, the same $K$ for training and test is recommended. $K = 6$ is enough for CD and NMF, while even $K = 1$ is acceptable for VQ. Typically 3∼6 steps are enough since simple MD's prior is still biased, and full convergence can overfit it. The few iterations are cheap and act as *early stopping*.

## 3.2   A CLOSE LOOK AT HAMBURGER

To understand the behavior of Hamburger in the networks, we visualize the spectrums of representations before and after Hamburger on the PASCAL VOC validation set. The input and output tensors are unfolded to $\mathbb{R}^{C \times HW}$. The accumulative ratio of squared largest $r$ singular values over total squared singular values of the unfolded matrix has been shown in Fig. 5. A truncated spectrum is usually observed in classic matrix decomposition models' results due to the low-rank reconstruction. In the networks, Hamburger also promotes energy concentration while preserving informative details via the skip connection. Additionally, we visualize the feature maps before and after Hamburger in Fig. 6. MD helps Hamburger learn interpretable global information by zeroing out uninformative channels, removing irregular noises, and completing details according to the context.

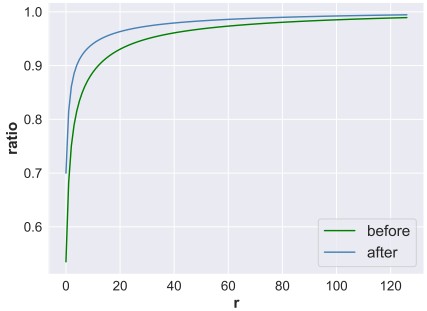

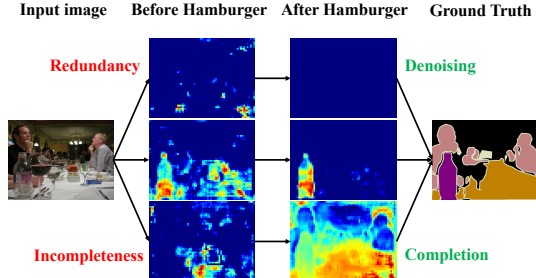

Figure 5: Accumulative Ratio          Figure 6: Visualization of feature maps

## 3.3   A COMPARISON WITH ATTENTION

This section shows the advantages of MD-based Hamburger over attention-related context modules in computational cost, memory consumption, and inference time. We compare Hamburger (Ham) with self-attention (SA) (Vaswani et al., 2017), Dual Attention (DA) module from DANet (Fu et al., 2019), Double Attention module from $A^2$ Net (Chen et al., 2018b), APC module from APCNet (He et al., 2019b), DM module from DMNet (He et al., 2019a), ACF module from CFNet (Zhang et al., 2019b), reporting parameters and costs of processing a tensor $\mathcal{Z} \in \mathbb{R}^{1 \times 512 \times 128 \times 128}$ in Tab. 3. Excessive memory usage is the key bottleneck of cooperating with attention in real applications. Hence we also provide the GPU load and inference time on NVIDIA TITAN Xp. In general, Hamburger is light in computation and memory compared with attention-related global context modules.

Table 3: Comparisons between Hamburger and context modules.

| Method | Params | MACs | GPU Load | | GPU Time | |
|---|---|---|---|---|---|---|
| | | | Train | Infer | Train | Infer |
| SA | 1.00M | 292G | 5253MB | 2148MB | 242.0ms | 82.2ms |
| DA | 4.82M | 79.5G | 2395MB | 2203MB | 72.6ms | 64.4ms |
| $A^2$ | 1.01M | 25.7G | 326MB | 165MB | 22.9ms | 8.0ms |
| APC | 2.03M | 17.6G | 458MB | 264MB | 26.5ms | 11.6ms |
| DM | 3.00M | 35.1G | 557MB | 268MB | 65.7ms | 23.3ms |
| ACF | 0.75M | 79.5G | 1380MB | 627MB | 71.0ms | 22.6ms |
| Ham (CD) | 0.50M | **16.2G** | **162MB** | 102MB | 20.0ms | 13.0ms |
| Ham (NMF) | 0.50M | 17.6G | 202MB | **98MB** | **15.6ms** | **7.7ms** |

## 3.4 SEMANTIC SEGMENTATION

We benchmark Hamburger on the PASCAL VOC dataset (Everingham et al., 2010), and the PASCAL Context dataset (Mottaghi et al., 2014), against state-of-the-art attentions. We use ResNet-101 (He et al., 2016) as our backbone. The output stride of the backbone is 8. The segmentation head is the same as ablation experiments. NMF is usually better than CD and VQ in ablation studies (see Tab. 1). Therefore, we mainly test NMF in further experiments. We use *HamNet* to represent ResNet with Hamburger in the following section.

Results on the PASCAL VOC test set, and the PASCAL Context validation set, are illustrated in Tab. 4, and Tab. 5, respectively. We mark all attention-based models with * in which diverse attentions compose the segmentation heads. Though semantic segmentation is a saturated task, and most contemporary published works have approximate performances, Hamburger shows considerable improvements over previous state-of-the-art attention modules.

Table 4: Comparisons with state-of-the-art on the PASCAL VOC test set w/o COCO pretraining.

| Method | mIoU(%) |
|---|---|
| PSPNet (Zhao et al., 2017) | 82.6 |
| DFN* (Yu et al., 2018) | 82.7 |
| EncNet (Zhang et al., 2018) | 82.9 |
| DANet* (Fu et al., 2019) | 82.6 |
| DMNet* (He et al., 2019a) | 84.4 |
| APCNet* (He et al., 2019b) | 84.2 |
| CFNet* (Zhang et al., 2019b) | 84.2 |
| SpyGR* (Li et al., 2020) | 84.2 |
| SANet* (Zhong et al., 2020) | 83.2 |
| OCR* (Yuan et al., 2020) | 84.3 |
| HamNet | **85.9** |

Table 5: Results on the PASCAL-Context Val set.

| Method | mIoU(%) |
|---|---|
| PSPNet (Zhao et al., 2017) | 47.8 |
| SGR* (Liang et al., 2018) | 50.8 |
| EncNet (Zhang et al., 2018) | 51.7 |
| DANet* (Fu et al., 2019) | 52.6 |
| EMANet* (Li et al., 2019) | 53.1 |
| DMNet* (He et al., 2019a) | 54.4 |
| APCNet* (He et al., 2019b) | 54.7 |
| CFNet* (Zhang et al., 2019b) | 54.0 |
| SpyGR* (Li et al., 2020) | 52.8 |
| SANet* (Zhong et al., 2020) | 53.0 |
| OCR* (Yuan et al., 2020) | 54.8 |
| HamNet | **55.2** |

## 3.5 IMAGE GENERATION

Attention presents as the global context description block in deep generative models like GANs. Most state-of-the-art GANs for conditional image generation integrate self-attention into their architectures since SAGAN (Zhang et al., 2019a), *e.g.,* BigGAN (Brock et al., 2018), S³GAN (Lučić et al., 2019), and LOGAN (Wu et al., 2019). It is convincing to benchmark MD-based Hamburger in the challenging conditional image generation task on ImageNet (Deng et al., 2009).

Table 6: Results on ImageNet 128×128. * are from Tab. 1 and Tab. 2 of Zhang et al. (2019a).

| Method | FID↓ |
|---|---|
| SNGAN-projection* | 27.62 |
| SAGAN* | 18.28 |
| HamGAN-baby | 16.05 |
| YLG | 15.94 |
| HamGAN-strong | 14.77 |

Experiments are conducted to compare Hamburger with self-attention on ImageNet 128×128. Self-attention is replaced by Hamburger with NMF ham in both generator and discriminator at feature resolution 32×32, named as *HamGAN-baby*. HamGAN achieves an appreciable improvement in Fréchet Inception Distance (FID) (Heusel et al., 2017) over SAGAN. Additionally, we compare Hamburger with a recently developed attention variant Your Local GAN (YLG) (Daras et al., 2020) using their codebase and the same training settings, named *HamGAN-strong*. HamGAN-strong offers over 5% improvement in FID while being 15% faster for the total training time and 3.6x faster for the module time (1.54 iters/sec of HamGAN, 1.31 iters/sec of YLG, and 1.65 iters/sec without both context modules, averaged from 1000 iterations) on the same TPUv3 training platform.

## 4 RELATED WORK

**General Survey for Attention** The last five years have witnessed a roaring success of attention mechanisms (Bahdanau et al., 2015; Mnih et al., 2014; Xu et al., 2015; Luong et al., 2015) in

deep learning. Roughly speaking, the attention mechanism is a term of adaptively generating the targets' weights to be attended according to the requests. Its architectures are diverse, and the most well-known one is dot product self-attention (Vaswani et al., 2017). The attention mechanism has a wide range of applications, from a single source (Lin et al., 2017) to multi-source inputs (Luong et al., 2015; Parikh et al., 2016), from global information discovery (Wang et al., 2018; Zhang et al., 2019a) to local feature extraction (Dai et al., 2017; Parmar et al., 2019).

Previous researchers attempt to explain the effectiveness of attention mechanisms from numerous aspects. Capturing long-range dependencies (Wang et al., 2018), sequentially decomposing visual scenes (Eslami et al., 2016; Kosiorek et al., 2018), inferring relationships between the part and the whole (Sabour et al., 2017; Hinton et al., 2018), simulating interactions between objects (Greff et al., 2017; van Steenkiste et al., 2018), and learning the dynamics of environments (Goyal et al., 2019) are often considered as the underlying mechanisms of attention.

One common idea from biology is that attention simulates the emergence of concerns in many unconscious contexts (Xu et al., 2015). Some work tries to interpret the attention mechanism by visualizing or attacking attention weights (Serrano & Smith, 2019; Jain & Wallace, 2019; Wiegreffe & Pinter, 2019), while others formulate attention into non-local operation (Wang et al., 2018) or diffusion models (Tao et al., 2018; Lu et al., 2019) or build attention-like models via Expectation Maximization (Greff et al., 2017; Hinton et al., 2018; Li et al., 2019) or Variational Inference (Eslami et al., 2016) on a mixture model. A connection between transformer and graph neural network is discussed as well (Liang et al., 2018; Zhang et al., 2019c). Overall, discussions towards attention are still far from reaching agreements or consistent conclusions.

**Efficient Attention**   Recent works develop efficient attention modules via low-rank approximation in both computer vision (Chen et al., 2018b; Zhu et al., 2019; Chen et al., 2019; Li et al., 2019) and natural language processing (Mehta et al., 2019; Katharopoulos et al., 2020; Wang et al., 2020; Song et al., 2020). Technically, the low-rank approximation usually targets at the correlation matrix, i.e., the product of $Q$ and $K$ after the $softmax$ operation, using a product of two smaller matrices to approximate the correlation matrix and applying the associative law to save the memory cost and computation, where the approximation involves kernel functions or other similarity functions. Other works (Babiloni et al., 2020; Ma et al., 2019) make efforts to formulate attention into tensor form but may generate large intermediate variables. In this paper, we do not approximate attention or make it efficient. This paper formulates modeling the global context as a low-rank completion problem. The computation and memory efficiency is a by-product of the low-rank assumption on the clean signal subspace and optimization algorithms as architectures.

**Matrix Decomposition in Deep Learning**   There is a long history of combining MD with deep learning. Researchers focus on reducing the parameters in the networks via factorization on the weights, including the softmax layer (Sainath et al., 2013), the convolutional layer (Zhong et al., 2019), and the embedding layer (Lan et al., 2019). Tariyal et al. (2016) attempts to construct deep dictionary learning for feature extraction and trains the model greedily. This paper tries to factorize the representations to recover a clean signal subspace as the global context and provide a new formulation for modeling the long-range dependencies via matrix decomposition.

## 5   CONCLUSION

This paper studies modeling long-range dependencies in the networks. We formulate learning the global context as a low-rank completion problem. Inspired by such a low-rank formulation, we develop the Hamburger module based on well-studied matrix decomposition models. By specializing matrix decomposition's objective function, the computational graph created by its optimization algorithm naturally defines ham, Hamburger's core architecture. Hamburger learns interpretable global context via denoising and completing its input and improves the spectrum's concentration. It is startling that, when prudently coped with the backward gradient, even simple matrix decomposition proposed 20 years ago is as powerful as self-attention in challenging vision tasks semantic segmentation and image generation, as well as light, fast, and memory-efficient. We plan to extend Hamburger to natural language processing by integrating positional information and designing a decoder like Transformer, build a theoretical foundation for the one-step gradient trick or find a better method to differentiate MDs, and integrate advanced MDs in the future.

ACKNOWLEDGMENTS

Zhouchen Lin is supported by NSF China (grant no.s 61625301 and 61731018), Major Scientific Research Project of Zhejiang Lab (grant no.s 2019KB0AC01 and 2019KB0AB02), Beijing Academy of Artificial Intelligence, and Qualcomm. We thank Google's Tensorflow Research Cloud (TFRC) for providing us Cloud TPUs.

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

## A    TABLE OF NOTION

Table 7: Summary of notations in this paper

| | |
|---|---|
| $\beta$ | A scalar. |
| $\mathbf{x}$ | A vector. |
| $\boldsymbol{X}$ | A matrix. |
| $\mathcal{Z}$ | A tensor. |
| $\mathbf{1}_n$ | A vector whose n elements are all 1. |
| $\mathbf{x}_i$ | i-th column of matrix $\boldsymbol{X}$. |
| $\mathbf{h}^t$ | Vector $\mathbf{h}$ at time step $t$. |
| $\frac{\partial \mathbf{y}}{\partial \mathbf{x}}$ | Jacobian matrix of $\mathbf{y}$ w.r.t. $\mathbf{x}$. |
| $\|\boldsymbol{X}\|$ | Operator norm. |
| $\|\boldsymbol{X}\|_F$ | Frobenius norm. |
| $diag$ | Map a vector to a diagonal matrix. |
| $cosine$ | Cosine similarity used in Alg. 1. |
| $softmax$ | Column-wise softmax function. |
| $normalize$ | Column-wise normalization by $L_2$ norm. |

## B    HAMS

Additionally, we introduce another type of ham adopted by Hamburger, Concept Decomposition.

**Concept Decomposition**    We first enhance Concept Decomposition (Dhillon & Modha, 2001) to the following form:

$$\min_{\boldsymbol{D},\boldsymbol{C}} \|\boldsymbol{X} - \boldsymbol{D}\boldsymbol{C}\|_F^2 + \beta\|\boldsymbol{C}\|_F^2$$
$$\text{s.t. } \boldsymbol{D} \in \arg\max_{\boldsymbol{D}} \mathcal{Q}(\boldsymbol{D}, \boldsymbol{X}). \tag{14}$$

This problem has a closed solution *w.r.t.* $\boldsymbol{C}$ under a given $\boldsymbol{D}$, *i.e.,* $\boldsymbol{C} = (\boldsymbol{D}^\top\boldsymbol{D} + \beta\boldsymbol{I})^{-1}\boldsymbol{D}^\top\boldsymbol{X}$. Since $\boldsymbol{D}^\top\boldsymbol{D} + \beta\boldsymbol{I}$ is a positive definite matrix with a regularized conditional number, the inverse can be more numerically stable than the original one where a semi-positive definite matrix $\boldsymbol{D}^\top\boldsymbol{D}$ is given under $\beta = 0$. In practice, 0.01 or 0.1 makes no difference for $\beta$.

---

**Algorithm 3** Ham: Soft CD

---

Input $\boldsymbol{X}$. Initialize $\boldsymbol{D}, \boldsymbol{C}$
**for** $k$ from 1 to $K$ **do**
  $\boldsymbol{C} \leftarrow softmax(\frac{1}{T}cosine(\boldsymbol{D}, \boldsymbol{X}))$
  $\boldsymbol{D} \leftarrow normalize(\boldsymbol{X}\boldsymbol{C}^\top)$
**end for**
$\boldsymbol{C} \leftarrow (\boldsymbol{D}^\top\boldsymbol{D} + \beta\boldsymbol{I})^{-1}\boldsymbol{D}^\top\boldsymbol{X}$
Output $\bar{\boldsymbol{X}} = \boldsymbol{D}\boldsymbol{C}$.

---

The dictionary in CD is given by spherical K-means (Dhillon & Modha, 2001) with objective $\mathcal{Q}(\boldsymbol{D}, \boldsymbol{X})$, as mentioned in Eq. (14).

$$\arg\max_{\boldsymbol{D},\{\pi_j\}_r} \sum_{j=1}^r \sum_{\mathbf{x}\in\pi_j} cosine(\mathbf{x}, \mathbf{d}_j)$$
$$\text{s.t.} \qquad \|\mathbf{d}_j\| = 1. \tag{15}$$

The same strategy as VQ is adopted to make the whole algorithm differentiable, however, in which each column of $\boldsymbol{D}$ is normalized to be a unit vector and thus differs from VQ.

## C  PROOF OF PROPOSITIONS

We investigate an abstract RNN model inspired by numerical methods to understand the drawbacks of BPTT algorithm in differentiating the optimization algorithm of MDs, $\mathcal{M}$. We show the propositions in Sec. 2.3 to illustrate the unstable gradient from $\mathcal{M}$ when using BPTT algorithm, considering MDs' nature as optimization algorithms.

**Proposition 1**  The iterations of $\mathcal{F}$ have linear convergence.

*Proof.* It is obvious that $\mathcal{F}$ is a contraction mapping *w.r.t.* $\mathbf{h}$ under arbitrary given $\mathbf{x}$. We can then conclude $\{\mathbf{h}^t\}$ is a Cauthy sequence and $\mathcal{F}(*, \mathbf{x})$ admits a unique fixed point $\mathbf{h}^*$ due to Banach Fixed Point Theorem.

$$\|\mathbf{h}^{t+1} - \mathbf{h}^*\| = \|\mathcal{F}(\mathbf{h}^t, \mathbf{x}) - \mathcal{F}(\mathbf{h}^*, \mathbf{x})\| \tag{16}$$
$$\leq L_h \|\mathbf{h}^t - \mathbf{h}^*\|$$

Eq. (16) shows the linear convergence.

**Proposition 2**  $\lim_{t \to \infty} \frac{\partial \mathbf{y}}{\partial \mathbf{x}} = \frac{\partial \mathbf{y}}{\partial \mathbf{h}^*}(\boldsymbol{I} - \frac{\partial \mathcal{F}}{\partial \mathbf{h}^*})^{-1}\frac{\partial \mathcal{F}}{\partial \mathbf{x}}$.

*Proof.* Note that $\mathcal{F}(*, \mathbf{x})$ admits a unique fixed point $\mathbf{h}^*$ under arbitrary given $\mathbf{x}$, *i.e.,*

$$\mathbf{h}^* = \mathcal{F}(\mathbf{h}^*, \mathbf{x}) \quad \implies \quad \mathbf{h}^* - \mathcal{F}(\mathbf{h}^*, \mathbf{x}) = \mathbf{0} \tag{17}$$

By differentiating the above equation, we can obtain

$$(\boldsymbol{I} - \frac{\partial \mathcal{F}}{\partial \mathbf{h}^*})\frac{\partial \mathbf{h}^*}{\partial \mathbf{x}} = \frac{\partial \mathcal{F}}{\partial \mathbf{x}} \tag{18}$$

The Jacobian matrix $\boldsymbol{I} - \frac{\partial \mathcal{F}}{\partial \mathbf{h}^*}$ is invertible, which implies the existence of the implicit function $\mathbf{h}^*(\mathbf{x})$. Immediately, we have

$$\lim_{t \to \infty} \frac{\partial \mathbf{y}}{\partial \mathbf{x}} = \frac{\partial \mathbf{y}}{\partial \mathbf{h}^*}\frac{\partial \mathbf{h}^*}{\partial \mathbf{x}} = \frac{\partial \mathbf{y}}{\partial \mathbf{h}^*}(\boldsymbol{I} - \frac{\partial \mathcal{F}}{\partial \mathbf{h}^*})^{-1}\frac{\partial \mathcal{F}}{\partial \mathbf{x}}, \tag{19}$$

which completes the proof.

**Proposition 3**  $\lim_{t \to \infty} \|\frac{\partial \mathbf{y}}{\partial \mathbf{h}^0}\| = 0, \lim_{t \to \infty} \|\frac{\partial \mathbf{y}}{\partial \mathbf{x}}\| \leq \frac{L_{\mathcal{G}} L_x}{1 - L_h}$.

*Proof.*

$$\|\frac{\partial \mathbf{y}}{\partial \mathbf{h}^0}\| = \|\frac{\partial \mathbf{y}}{\partial \mathbf{h}^t}\prod_{i=1}^{t}\frac{\partial \mathbf{h}^i}{\partial \mathbf{h}^{i-1}}\| \leq \|\frac{\partial \mathbf{y}}{\partial \mathbf{h}^t}\|\prod_{i=1}^{t}\|\frac{\partial \mathbf{h}^i}{\partial \mathbf{h}^{i-1}}\| \leq L_{\mathcal{G}} L_h^t \tag{20}$$

Then we have:

$$\lim_{t \to \infty} \|\frac{\partial \mathbf{y}}{\partial \mathbf{h}^0}\| = 0. \tag{21}$$

$$\|\frac{\partial \mathbf{y}}{\partial \mathbf{x}}\| = \|\sum_{i=0}^{t}\frac{\partial \mathbf{y}}{\partial \mathbf{h}^t}\prod_{j=i+1}^{t}\frac{\partial \mathbf{h}^j}{\partial \mathbf{h}^{j-1}}\frac{\partial \mathbf{h}^i}{\partial \mathbf{x}}\|$$
$$\leq \sum_{i=0}^{t}\|\frac{\partial \mathbf{y}}{\partial \mathbf{h}^t}\|\prod_{j=i+1}^{t}\|\frac{\partial \mathbf{h}^j}{\partial \mathbf{h}^{j-1}}\|\|\frac{\partial \mathbf{h}^i}{\partial \mathbf{x}}\|$$
$$\leq L_{\mathcal{G}}(\sum_{i=0}^{t-1}L_h^i)L_x \tag{22}$$
$$= \frac{L_{\mathcal{G}} L_x(1 - L_h^t)}{1 - L_h}$$

Then we have:

$$\lim_{t \to \infty} \|\frac{\partial \mathbf{y}}{\partial \mathbf{x}}\| \leq \frac{L_{\mathcal{G}} L_x}{1 - L_h}. \tag{23}$$

## D    DATASETS

**PASCAL VOC**    The PASCAL VOC dataset (Everingham et al., 2010) is a widely used dataset in both semantic segmentation and detection. For segmentation, it contains 10,582 images for training, 1,449 images for validation and 1,456 images for testing. PASCAL VOC dataset involves 20 foreground object classes and a background class for segmentation and detection.

**PASCAL Context**    The PASCAL Context dataset (Mottaghi et al., 2014) is a challenging dataset in semantic segmentation, which provides detailed labels and involves 59 foreground object classes and a background class for segmentation. It consists of 4,998 and 5,105 images in training and validation set, respectively.

**ILSVRC 2012**    The ILSVRC 2012 (ImageNet) (Deng et al., 2009) dataset contains 1.3M training samples and 50k test images, categorized into 1000 object classes. We resize images to resolution $128 \times 128$, as done in SNGAN with projection (Miyato & Koyama, 2018) and SAGAN (Zhang et al., 2019a).

## E    DETAILS OF EXPERIMENTS

### E.1    ABALATION EXPERIMENTS

We use dilated ResNet-50 (He et al., 2016) with the output stride 16 as the backbone. The backbone is pre-trained on ImageNet (Deng et al., 2009). We apply a poly-learning rate policy under batch size 12 and 30k iterations (about 35 epochs) for fast experiments (less than 12 hours using 1 NVIDIA TITAN Xp GPU). The initial learning rate is set to 0.009, multiplied by $(1 - \frac{iter}{iter_{max}})^{0.9}$ after each iteration, with momentum 0.9 and weight decay 0.0001. Hyperparameters of Hamburger are the same as Appendix E.3.

### E.2    A COMPARISON WITH ATTENTION MECHANISM

We report MACs according to Molchanov et al. (2016), using torchprofile[1], a more accurate profiler for Pytorch. Real-time cost is measured by built-in Pytorch memory tools on NVIDIA TITAN Xp GPU with a input tensor $\mathcal{Z} \in \mathbb{R}^{1 \times 512 \times 128 \times 128}$. Inference times are averaged results from 20 repeats of 100 runs.

### E.3    SEMANTIC SEGMENTATION

**Architectures**    We use ResNet-101 (He et al., 2016) with the ouptput strid 8 as our backbone. We adopt dilated convolution (Chen et al., 2018a) to preserve more detail spatial information and enlarge receptive field as done in the backbone of state-of-the-art attention models (Fu et al., 2019; Li et al., 2019; Zhang et al., 2019b). We employ a $3 \times 3$ convolution layer with BN and ReLU to reduce channels from 2048 to 512 and then add Hamburger on the top of the backbone. Note that the input of Hamburger is a tensor $\mathcal{Z} \in \mathbb{R}^{C \times H \times W}$. We unfold $\mathcal{Z}$ to a matrix $\boldsymbol{Z} \in \mathbb{R}^{C \times HW}$ and set $d_z = C$ and $n = HW$ for Hamburger. Latent dimension $d$ and $r$, *i.e.,* the column vectors' dimension of the input matrix $\boldsymbol{X} \in \mathbb{R}^{d \times n}$ to $\mathcal{M}$ and the number of atoms in the dictionary $\boldsymbol{D} \in \mathbb{R}^{r \times d}$, are set to 512 and 64. The iterations of MD's optimization algorithm, $K$, are set to 6. Non-negative Matrix Factorization (NMF) is our default ham for semantic segmentation.

**Data augmentation**    In the training stage, we apply random left-right flipping, random scaling (from 0.5 to 2), and cropping to augment the training data. Images are resized to $513 \times 513$ for the PASCAL VOC dataset and the PASCAL Context dataset. In the test stage, the multi-scale and flipping strategy is applied as other state-of-the-art attention-based models (Fu et al., 2019; Yuan & Wang, 2018; Yuan et al., 2020).

---

[1] https://github.com/zhijian-liu/torchprofile

**Optimization**    We use mini-batch SGD with momentum 0.9 to train HamNet. Synchronized Batch Normalization is adopted in experiments on semantic segmentation. All backbones are fine-tuned from ImageNet (Deng et al., 2009) pre-training. Following previous works (Zhao et al., 2017; Chen et al., 2018a), we apply a poly-learning rate policy. The initial learning rate is multiplied by $(1 - \frac{iter}{iter_{max}})^{0.9}$. For the PASCAL VOC dataset, learning rate, weight decay, batch size, iterations are set to 0.009, 0.0001, 16, and 60k, respectively. We fine-tune HamNet on the PASCAL VOC trainval set with the learning rate down to a tenth. The learning rate, weight decay, batch size, iterations are 0.002, 0.0001, 16, and 25k for the PASCAL-Context dataset.

### E.4    IMAGE GENERATION

We use the official GAN codebase[2] from Tensorflow (Abadi et al., 2016) and TF-GAN to train HamGAN and evaluate FID.

**Architectures**    Experiments on ImageNet are conducted using the same architecture as SAGAN (Zhang et al., 2019a), and YLG (Daras et al., 2020), including Spectral Normalization (Miyato et al., 2018) in both the generator and the discriminator, conditional Batch Normalization in the generator, and class projection in the discriminator (Miyato & Koyama, 2018). Hamburger with NMF ham is placed at feature resolution 32×32 in both the generator and the discriminator where self-attention can obtain the best FID according to Zhang et al. (2019a). We use $d = 8r$ for Hamburger, and $d$ is the same as the input channels, while the optimization steps $K$ are 6. Restricted to expenditures of training GANs on ImageNet, $d$, $r$, and $K$ are decided according to the ablation experiments on semantic segmentation without new ablation experiments.

**Optimization**    For all models, we use Adam (Kingma & Ba, 2015) optimizer with TTUR (Heusel et al., 2017). HamGAN employs the same training settings as SAGAN (Miyato et al., 2018) and YLG (Daras et al., 2020), respectively.

**Evaluation metrics**    The quality of images generated by GANs are evaluated by Fréchet Inception Distance (FID) (Heusel et al., 2017). Lower FID indicates that the model can generate higher-fidelity images. In our experiments, 50k images are sampled from the generator to compute FID. We evaluate HamGAN for 6 runs and report the best FID to approximately match the convention in the modern GAN research like Kurach et al. (2019) and CR-GAN (Zhang et al., 2020), reporting top 5%/15% results in the experiments.

## F    FURTHER RESULTS FROM ABLATION EXPERIMENTS

Table 8: Ablation on initializations.

| Init | NMF | CD | VQ |
|------|------|------|------|
| fixed | 77.4(77.3) | 77.7(77.4) | 77.3(76.9) |
| learned | 76.8(76.5) | 75.0(73.7) | 75.9(75.8) |
| random | **78.3(77.8)** | 77.9(77.3) | 77.7(**77.4**) |
| online | 77.8(77.5) | **78.1(77.5)** | **78.0**(77.2) |

**Initialization**    We test four types of initialization for the dictionary $D$, including fixed initialization, learned initialization, random initialization, and warm start with online update. Usually, random initialization is the best choice that means we can sample each entry of $D$ from a given distribution like Uniform$(0, 1)$ as the initialization of the optimization algorithm $\mathcal{M}$. For NMF, after initializing $D$, we initialize $C = softmax(\frac{1}{T}cosine(D, X))$ since K-means is usually applied for initializing NMF and this initialization for $C$ is equivalent to a single update in Spherical K-means. A special reminder is that it is not suitable to initialize either $D$ or $C$ to values too close to 0 due to the property of the MU rule. So the temperature $T$ is recommended to be a higher value like 1 in this

---

[2]https://github.com/tensorflow/gan

initialization for $C$. Random initialization also works for $C$ in NMF with scores 77.8(77.6) when sampling $C_{ij} \sim \text{Uniform}(0, 1)$. Note that learned initialization is always the worst one since the BPTT algorithm is employed to learn the initialization that the gradient from $\mathcal{M}$ may impede the training of the backbone, instead of the one-step gradient. Warm start benefits MD with unit vectors in the dictionary $D$ like CD. In general, random initialization is good enough for all three selected MD models. A possible reason is that it can enforce the network to adapt to the results solved by different initializations during the training process, acting like an *inner augmentation*.

**Temperature** $T$    As we have claimed, when $T$ approaches 0, we can get a solution close to the original problem in both VQ and CD. In VQ and CD experiments, a relatively low temperature $T$ is more recommended to solve a better $D$ for MD. However, it will not receive more gains but increase the variance during training if we further lower $T$.

Table 9: Influence of temperature $T$ with CD ham.

| Temperature $T$ | mIoU(%) |
|---|---|
| 1 | 77.1(77.0) |
| 0.1 | 78.2(77.5) |
| 0.01 | 78.1(77.5) |

**Iterations** $K$    We take the iterations $K$ of optimization algorithms $\mathcal{M}$ for all three MD models, NMF, CD, and VQ, into our consideration. More iterations and even fully converged results for $\mathcal{M}$ are tested in the evaluation stage but worse than little optimization steps. The smaller $K$, ranging from 1 to 8, can be treated as *early stopping* for the optimization algorithm $\mathcal{M}$, obtaining satisfactory performances. For a detailed visualization, see Fig. 7, Fig. 8, Fig. 9.

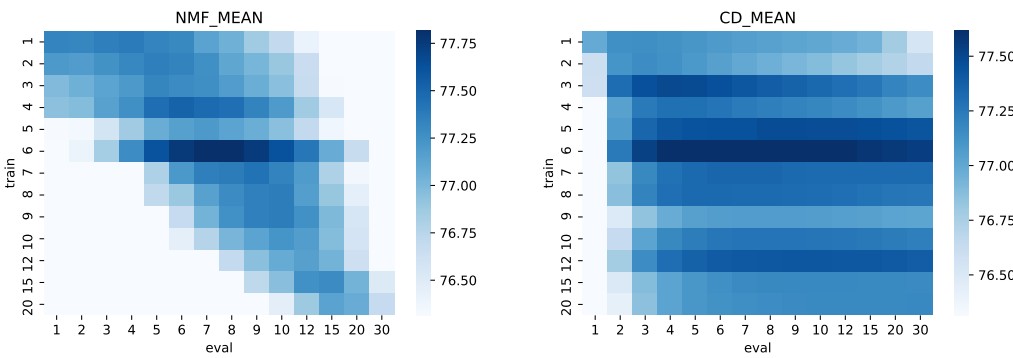

Figure 7: Impacts of $K$ on NMF          Figure 8: Impacts of $K$ on CD

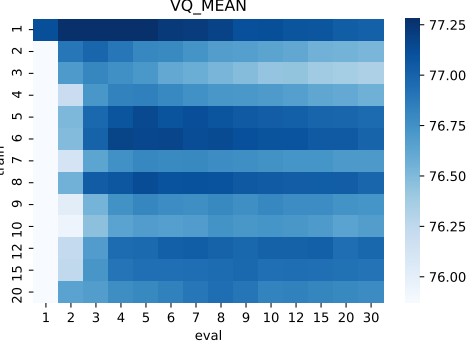

Figure 9: Impacts of $K$ on VQ

## G    AN INTUITIVE ILLUSTRATION

In this section, we hope to give an example to help our readers develop insight into why the low-rank assumption is useful for modeling the representations' global context.

The low-rank assumption helps because it represents the inductive bias that *the low-level representations contain limited and much less high-level concepts than the scale of the representations themselves*. Imagine an image in which a person works on the road. Many hyper-pixels extracted by the backbone CNN will describe the road. Note that the road can be considered as repeats of small road patches, which means that we can represent the road via modeling the basic road patches and repeat them. Mathematically, it is equivalent to finding a small set of bases $D$ corresponding to different road patches and a coefficient matrix $C$ that captures the relation between the elementary road patches and the hyper-pixels. This example illustrates that the high-level concepts, *i.e.,* the global context, can be low-rank in the ideal situation.

The hyper-pixels describing the road patches have close semantic attributes. However, due to the vanilla CNN's inefficiency for modeling the long-range dependencies, the learned representation contains too many local details and incorrect information, lacking global guidance. Imagine that the person in the image wears gloves. When we see the gloves patch locally, we think that this patch describes gloves. When we consider the global context, we can understand that this patch is a part of a person. The semantic information is hierarchical, depending on at which level we hope to comprehend. This work aims at enabling the networks to understand the context globally via the low-rank completion formulation. We thus model the incorrect information, namely the redundancies and incompleteness, as a noise matrix. To emphasize the global context, we decompose the representations into two parts, a low-rank global information matrix and a noise matrix, by employing the optimization algorithm to recover the clean signal subspace, discard the noises, and enhance the global information via the skip connection. It could be learned from the data on how much global information the networks need for a specific task.

