# OpenReview forum: "Is Attention Better Than Matrix Decomposition?"
_ICLR.cc/2021/Conference — ICLR 2021 Poster_

### Official Review · AnonReviewer1 · 2020-10-27
**MD could be an alternative for "self"-attention.**

**Rating:** 6
**Confidence:** 3

**Review:**

Summary
1. This paper finds that matrix decomposition (MD) performs well as well as the self-attention.
2. According to the paper, MD approximate the given matrix with low-rank, and it might be helpful inductive bias
3. MD can be implemented with vanilla matrix factorization or non-negative matrix factorization.
4. For stable learning, this paper proposes an additional technique, one-step gradient instead of back-propagation through time (BPTT)
5. To validate the performance, they experiment on semantic segmentation and image generation.

Please reply to the below questions.
1. What is the advantage of MD over attention? According to the paper, the main advantage of MD is a less computational burden, but there are several works for linear time attention. Furthermore, it is hard to find sufficient reasons that MD performs better than attention-based models.
2. There is no description for Table 1. Could you explain the table 1?
3. Are there results for MD with BPTT instead of a one-step gradient on segmentation or image generation task?
4. What is the "human priors" in the conclusion section? Does it denote the "low-rank"?
It is hard to understand that low-rank will be helpful.
If MD set r as same as min(d,n) instead of small r, does MD have lower performance?
5. There are several works about 1) analyzing the low-rank problems in multi-head attention and 2) incorporating the low-rank approximation into attention. The discussion between this paper and related works is not enough.
6. The title is "Is Attention Better Than Matrix Decomposition?", but the paper is only for the "self"-attention and MD.
Are there results for Encoder-Decoder structured tasks (such as Translation)?

I suggest that this paper discusses the relationship between MD and the below papers.

a) Factorization and Attention
1. A Tensorized Transformer for Language Modeling

b) Low-rank problems and attention
1. Low-Rank Bottleneck in Multi-head Attention Models

c) Low-rank attention
1. Transformers are rnns: Fast autoregressive transformers with linear attention
2. Linformer: Self-Attention with Linear Complexity
3. Implicit Kernel Attention
4. Compact Multi-Head Self-Attention for Learning Supervised Text Representations

-----
I read a valuable author's response, and I keep my positive score.

---

> ### Author Response · Authors · 2020-11-24
> **Response to R1 (1/3)**
>
> We sincerely thank Reviewer 1 for understanding this work and evaluating it from the perspective of the natural language processing community. In this response, we hope to answer the questions and address the concerns on why the low-rank inductive bias helps to model the global context and why MD works at a small $r$. Discussion with attention modules in natural language processing is considered as well.
>
> > What is the advantage of MD over attention?
> >
> > It is hard to find sufficient reasons that MD performs better than attention-based models.
>
> **A1**: The advantages of MD-based Hamburger over attention-related methods for modeling the global context in computer vision are at least threefolds.
>
> 1. We formulate the networks' global context issues as a low-rank completion problem of the input tensor. This optimization problem is a new general formulation for modeling the global context, and a unified framework to ***derive*** context modules ***in batch*** by specializing the objective function and developing the optimization algorithm. Based on this formulation, we further consider integrating the solution to the low-rank completion problem, i.e., the optimization algorithm to solve MD, as the architectural elements. Moreover, if the formulation is valid, the derived MD-based modules can *all* work for modeling the global context in computer vision, which supports the potential of the formulation itself.
>
> 2. Based on the formulation for modeling the global context, MD-based Hamburger can show *appreciable improvements* over visual attention modules in semantic segmentation and image generation regarding the performance. Semantic segmentation is a saturated task where attention-related segmentation heads or multi-scale adaptive context methods govern this field since 2018. Published works since 2019 have similar performances ranging from 84.2 to 84.4 on the PASCAL VOC test set [1,2,3,4,5]. In this paper, Hamburger-equipped HamNet pushes the state-of-the-art records to 85.9 mIoU, which validates the superiority of both the formulation for modeling the global context as the low-rank completion problem and architecture design from the optimization algorithms.
>
> 3. The memory and computational cost for modeling the global context is visibly lower than commonly used visual attention modules. Note that the literature in computer vision rarely claims the linear complexity but compares the real cost since the scale of the feature maps is quadratic w.r.t. the length of the image; thus, if a visual attention module is not of applicable complexity, it can be impractical to use such context modules in dense prediction tasks like semantic segmentation.  Hamburger benefits in computation from the low-rank assumption on the latent structures.
>
> > Could you explain the table 1?
> >
> > Are there results for MD with BPTT instead of a one-step gradient on segmentation or image generation task?
>
> **A2**: Sorry for missing the caption of Tab. 1. Tab. 1 compares the one-step gradient and BPTT algorithm on semantic segmentation with the ablation settings. The left column is mIoU scores for Hamburger under the one-step gradient. The right column is mIoU under the BPTT algorithm.

---

> > ### Author Response · Authors · 2020-11-24
> > **Response to R1 (2/3)**
> >
> > > What is the "human priors" in the conclusion section? Does it denote the "low-rank"? It is hard to understand that low-rank will be helpful. If MD set r as same as min(d,n) instead of small r, does MD have lower performance?
> >
> > **A3**: To describe the prior more formally, we use the statement in Sec. 2.2, i.e., ''In the ideal situation, the global information hidden in the representations can be low-rank.''. The low-rank assumption helps because it represents the inductive bias that *the low-level representations contain limited and much less high-level concepts than the scale of the representations themselves.* Imagine an image in which a person works on the road. Many hyper-pixels extracted by the backbone CNN will describe the road. Note that the road can be considered as repeats of small road patches, which means that we can represent the road via modeling the atom road patches and repeat them. Mathematically, it is equivalent to finding a small set of bases corresponding to different road patches and a coefficient matrix that captures the relation between the basic road patches and the hyper-pixels. The hyper-pixels describing the road patches have close semantic attributes. However, due to the vanilla CNN's inefficiency for modeling the long-range dependencies, the learned representation contains too many local details and incorrect information, lacking global guidance (Imagine that the person in the image wears gloves. When we see the gloves patch locally, we think that this patch describes gloves. When we consider the global context, we can understand that this patch is a part of a person. The semantic information is hierarchical, depending on at which level we hope to comprehend. This work aims at enabling the networks to understand the context globally via the low-rank completion formulation.). We thus model the redundancies and incompleteness as a noise matrix. To emphasize the global context, we decompose the representations into two parts, a low-rank global information matrix and a noise matrix, by employing the optimization algorithm to recover the clean signal subspace, and discard the noises and enhance the global information via the skip connection. It could be learned from the data on how much global information the networks need for a specific task.
> >
> > The dimension of the signal subspace, $r$, is an estimation on the *maximum capacity* and the structure of the global information, also a hyper-parameter in the current work. Note that using a larger $r$ does not mean that the recovered low-rank matrix must be of rank $r$. It is an *upper bound* of the output matrix's rank. Setting $r$ to $\min(d,n)$, e.g., $r=d=512$, and $K=6$ for NMF, can achieve 77.9(77.6) with ablation settings while $r=64$ can obtain 78.3(77.8) at several times less computation.

---

> > > ### Author Response · Authors · 2020-11-24
> > > **Response to R1 (3/3)**
> > >
> > > > There are several works about 1) analyzing the low-rank problems in multi-head attention and 2) incorporating the low-rank approximation into attention. The discussion between this paper and related works is not enough.
> > >
> > > **A4**: Thanks for the suggestions. We add the discussion to the related works, including the suggested papers. The critical point is that we do not think that we build an attention module. **MD is not attention.** Matrix decomposition and the low-rank completion problem emerged much earlier than attention modules. This work makes efforts towards giving the global context issue a proper formulation via these techniques. Under such a formulation, we employ MD and its optimization algorithm as the architectural elements. To verify this formulation, we compare Hamburger with visual global context modules, in which attention is the most famous and powerful one. We do not approximate attention or analyze why attention works, but build a framework to model the global context mathematically, as an alternative of attention families in the competitive vision tasks.
> > >
> > > Technically, the low-rank approximation of attention usually targets on the correlation matrix, i.e., the product of $\boldsymbol{Q}$ and $\boldsymbol{K}$ after the $softmax$ operation, using a product of two smaller matrices to approximate this correlation matrix and applying the associative law of matrix algebra to save the memory cost and computation, where the approximation involves kernel functions or other similarity functions, like Double Attention in $A^2$ Net [6] in computer vision and ''Transformers are RNNs'' [7] in natural language processing. As for the low-rank bottleneck in multi-head attention, Bhojanapalli, Srinadh et al. [8], and Shazeer, Noam, et al.[9] show different opinions. The former increases the key size to overcome the ''low-rank bottleneck'', while the latter reduces the key size while increasing the head number. Besides, the Tensorized Transformer [10] reformulates the entire attention function to the tensor form and learns the bases via projection. A special reminder is that Tensorized Transformer generates a tensor $\mathcal{X} \in \mathbb{R}^{n\times n \times d}$ according to the official code base, which is not acceptable for vision tasks (Consider $n=64*64$ for semantic segmentation). In contrast, this work formulates the global context as a low-rank completion problem and solves the low-rank reconstruction via the flexible optimization algorithm. The optimization algorithm is a strong prior for the architecture to solve the dictionary and code without learning from data.
> > >
> > > > The title is "Is Attention Better Than Matrix Decomposition?", but the paper is only for the "self"-attention and MD. Are there results for Encoder-Decoder structured tasks?
> > >
> > > **A5**: As current work does not generalize this method to natural language processing yet, we only claim that MD-based Hamburger is better than attention modules in semantic segmentation and conditional image generation via GANs for modeling the global context, rather than in other tasks or for other usages. In these tasks, attention-related context modules are in the form of self-attention. MD only receives one input matrix $\boldsymbol{X}$ in the context of this paper rather than multiple inputs for retrieving information, e.g., $\boldsymbol{Q}$, $\boldsymbol{K}$, and $\boldsymbol{V}$ of classic attention in the decoder of Transformer. Hence it is not applicable for the encoder-decoder task now. However, *as the future works listed in the Conclusion section*, we plan to extend Hamburger to natural language processing by incorporating the position information and designing a decoder like Transformer.
> > >
> > > [1] He, J. et al. “Dynamic Multi-Scale Filters for Semantic Segmentation.” 2019 IEEE/CVF International Conference on Computer Vision (ICCV) (2019): 3561-3571.
> > >
> > > [2] He, J. et al. “Adaptive Pyramid Context Network for Semantic Segmentation.” 2019 IEEE/CVF Conference on Computer Vision and Pattern Recognition (CVPR) (2019): 7511-7520.
> > >
> > > [3] Zhang, H. et al. “Co-Occurrent Features in Semantic Segmentation.” 2019 IEEE/CVF Conference on Computer Vision and Pattern Recognition (CVPR) (2019): 548-557.
> > >
> > > [4] Li, Xia et al. “Spatial Pyramid Based Graph Reasoning for Semantic Segmentation.” 2020 IEEE/CVF Conference on Computer Vision and Pattern Recognition (CVPR) (2020): 8947-8956.
> > >
> > > [5] Yuan, Y. et al. “Object-Contextual Representations for Semantic Segmentation.” ArXiv abs/1909.11065
> > >
> > > [6] Chen, Y. et al. “A2-Nets: Double Attention Networks.” ArXiv abs/1810.11579
> > >
> > > [7] Katharopoulos, Angelos et al. “Transformers are RNNs: Fast Autoregressive Transformers with Linear Attention.” ArXiv abs/2006.16236
> > >
> > > [8] Bhojanapalli, Srinadh et al. “Low-Rank Bottleneck in Multi-head Attention Models.” ArXiv abs/2002.07028
> > >
> > > [9] Shazeer, Noam et al. “Talking-Heads Attention.” ArXiv abs/2003.02436
> > >
> > > [10] Ma, X. et al. “A Tensorized Transformer for Language Modeling.” ArXiv abs/1906.09777

---

> > > ### Comment · AnonReviewer3 · 2020-11-24
> > > **Interesting and insightful example**
> > >
> > > I find the example given here particularly interesting as it provides useful insight on the proposed method. I think that the example (or maybe a more "formal" version of it) should be included in the manuscript also.

---

> > > > ### Author Response · Authors · 2020-11-24
> > > > **Thanks for the suggestion**
> > > >
> > > > We are glad to receive the kudos on this example. Indeed we like it too. It is the prototype that motivates us to give such a formulation. However, we think it is not ''formal'' enough to include in the main text, so we do not. As there are only 2.5 hours to the rebuttal deadline and the revision has finished, and there is no room for the main text, maybe we can add this example to the appendix of this paper?

---

> > > > ### Author Response · Authors · 2020-11-25
> > > > **Update**
> > > >
> > > > Finally, we add this example to the appendix and mention it in the main text. We hope that it could help readers further understand this paper. Many thanks!

---

### Official Review · AnonReviewer2 · 2020-10-27
**overall this is a good submission**

**Rating:** 7
**Confidence:** 4

**Review:**

This paper proposes to use matrix decomposition to construct low-rank representations to find the long-distance correlations in context, which is demonstrated more effective than popular self-attention mechanism. Combining linear transformation and matrix decomposition (core part), authors design Hamburger block to model global dependencies from input as residual output. The authors propose differentiable modified Vector Quantization and Non-negative Matrix Factorization to perform matrix decomposition. They propose one-step gradient, an approximation of Back-Propagation Through Time (BPTT) algorithm, to back-propagate gradient of matrix decomposition. They conduct experiments on semantic segmentation and image generation to demonstrate the superiority of their methods regarding modelling global dependencies and computational cost.

I believe that there is clear novelty in the proposed method. The paper is well written. One weakness is that the experiment analysis is a little weak. It will be great to see stronger experiments in the final.

---

> ### Author Response · Authors · 2020-11-24
> **Response to R2**
>
> We sincerely thank Reviewer 2 for affirming the novelty and clarity of this paper. We indeed provide stronger results in this revision. We add systematic comparison with global context modules in computer vision concerning the computational cost. Also, we improve the score on Pascal VOC test set from 85.4 to 85.9, a new state-of-the-art result. The GAN part is enhanced as well. We evaluate HamGAN-strong for 6 runs and report the best FID 14.77 we obtain (the former is 14.88) to approximately match the convention in the modern GAN research like Kurach et al.[1] and CR-GAN[2], reporting top 5%/15% results in their experiments.
>
> [1] Kurach, Karol et al. “A Large-Scale Study on Regularization and Normalization in GANs.” ICML (2019).
>
> [2] Zhang, H. et al. “Consistency Regularization for Generative Adversarial Networks.” ArXiv abs/1910.12027 (2020)

---

### Official Review · AnonReviewer3 · 2020-10-28
**Matrix decomposition can help to capture global context more efficiently and with less overhead with respect to self-attention**

**Rating:** 8
**Confidence:** 4

**Review:**

# Summary:
The paper presents a method based on matrix decomposition (MD) for encoding global context in computer vision tasks. In particular, a "Hamburger" block is proposed encompassing matrix decomposition as its central part, between two linear projection layers. Direct comparison and relations are drawn between the proposed method and the widely adopted self-attention paradigm. The proposed method leads to improved results when Hamburger blocks are used instead of self-attention blocks, leading at the same time to reduced number of parameters, memory footprint and inference time.


# Strengths:
The paper presents a novel and simple method for capturing global context, demonstrated on two challenging computer vision tasks, namely semantic segmentation and image generation. Important advantages of the proposed method, with respect to the self-attention method which is usually employed in such tasks, are the fact that it can be easily adapted to a wide range of models and problems, it is more efficient and has reduced memory requirements.

A one-step gradient method is proposed for propagating the gradients through the MD optimization algorithm during training. One-step gradient is shown to overcome the unstable gradient problems of the back-propagation through time (BPTT) algorithm, leading to improved performance. A detailed analysis is provided comparing the two methods (one-step vs BPTT).

The evaluation is quite comprehensive comparing the proposed hamburger block with respect to similar self-attention blocks under several aspects (accuracy/FID score, GPU load, GPU time, FLOPs, nr. of parameters). A detailed ablation study is also presented showing the effect of the most important factors of the proposed contribution in the final performance.

Regarding writing quality, the paper is clear and easy to read. The main ideas and contributions are clearly stated and presented. Some issues regarding the structure of the sections is discussed below.

# Weaknesses:
The paper makes the more general claim that the proposed approach can be used for including any human inductive bias expressed through an optimization problem, however, only the problem of capturing global context, in place of self-attention, is explored. To support this more general claim it would be important to include some representative examples (even without providing a detailed evaluation on those).

Possibly related to the previous point is the observation that non-negative matrix factorization (NMF) seems to always perform better than the other two matrix decomposition methods, Vector Quantization (VQ) and Concept Decomposition (CD). This leads to certain questions, as for example:
* are there any problems that lend themselves better to the other types of MD?
* is the performance of VQ and CD degraded because they are rendered soft?
* what is the divergence of soft with original MD as far as the original optimization problem is concerned? The results of the ablation on temperature T provided in Appendix G partially show that the "softening" of the algorithm might negatively affect the accuracy.

I find it also strange and possibly nearly violating formatting that the related work section is provided in the appendix. Some directly related work is discussed in the main text, yet a more detailed discussion considering a broader set of works only appears in the appendix. Also regarding related work, other works employing matrix decomposition in the context of deep learning, are not covered (e.g. (Sainath et al., 2013; Tariyal et al., 2016)).

Sainath, T. N., Kingsbury, B., Sindhwani, V., Arisoy, E., & Ramabhadran, B. (2013). Low-rank matrix factorization for deep neural network training with high-dimensional output targets. In 2013 IEEE international conference on acoustics, speech and signal processing (pp. 6655-6659).

Tariyal, S., Majumdar, A., Singh, R., & Vatsa, M. (2016). Deep dictionary learning. IEEE Access, 4, 10096-10109.

## Minor Comments
* Table 1, no details are provided for the metric used for the results
* Table 6, it would be better to specify the difference between the two entries of HamGAN
* The text in almost all figure is quite small and very hard to read on typical zoom factors (~100%).

# Rating Justification
Overall, I think that the idea of using matrix decomposition as architectural element to capture global context is quite interesting and novel. Also the method shows advantages with respect to self-attention as far as efficiency and memory requirements are concerned. There are some issues regarding the generality of the proposed approach and the paper's structure, however, I think that the paper strengths exceed its weaknesses.

# Rating and comments after the rebuttal
The authors addressed my concerns in their feedback and the revised manuscript they have provided. In particular, I find the claimed contributions much clearer now. In my view, they have also suitably addressed the concerns raised in the other reviews. As a result, I increase my rating to 8 as I think that this work is interesting, novel and impactful.

---

> ### Author Response · Authors · 2020-11-24
> **Response to R3**
>
> We sincerely thank Reviewer 3 for comprehensively understanding this paper and evaluating it from many aspects. The proposed method's novelty, simplicity, and efficiency are affirmed by R3, while the concerns focus on this paper's structure and the generality of a claim. The following replies endeavor to address these concerns.
>
> ---
> > The paper makes the more general claim that the proposed approach can be used for including any human inductive bias expressed through an optimization problem.
>
> **A1**: Note that we do not claim that this approach can be applied to *any human inductive bias*. When this paper writes about the optimization-driven designing approach, we add a particular domain restricted to the global context issue. Though we agree that this approach (given a prior, formulate it into an optimization problem and convert the algorithm into an architecture) is possibly able to generalize to other priors in the architecture design, it is still too strong and too wide to claim it in the paper directly (actually we do not do so), because it is **highly non-trivial** to propose a proper inductive bias and formulate it into an objective function. We rephrase the claims in the revised version to make it more appropriate.
>
> > The observation that non-negative matrix factorization (NMF) seems to always perform better than the other two matrix decomposition methods, Vector Quantization (VQ) and Concept Decomposition (CD). This leads to certain questions, as for example:
> >
> > - are there any problems that lend themselves better to the other types of MD?
> > - is the performance of VQ and CD degraded because they are rendered soft?
> > - what is the divergence of soft with original MD as far as the original optimization problem is concerned? The results of the ablation on temperature T provided in Appendix G partially show that the "softening" of the algorithm might negatively affect the accuracy.
>
> **A2**: We hope to address several questions jointly since they are related intrinsically.
>
> The first point is why we choose these MD models. We hope to validate the generality of the proposed approach in modeling the global context, i.e., various MD models can *all* work when dealing with the gradient carefully. Hence we choose several MD models rather than only one model. It will be convincing that even those simplest MD models (regarding their proposed time and the complexity of the model itself) can be powerful enough to compare with state-of-the-art attention modules for encoding the long-range correlation in the highly competitive vision tasks. So we choose VQ, CD, and NMF, three simple, lightweight MD models proposed 20 years ago and famous enough (at least over 1000 citations) to verify this approach.
>
> The second point is crucial. It is about what makes an MD model perform better than others for modeling the global context in this paper. ***The differences among these models in performance might depend on which objective function models the prior we want the best, e.g., recovering the latent structure of the input tensor in this paper, the quality of the solution from the corresponding optimization algorithm, and which optimization algorithm is more friendly to the backward gradient.*** According to this paper, NMF performs better than VQ and CD in the given tasks and datasets, perhaps because NMF models the latent structure better than VQ and CD since the representations in the classic backbones are usually non-negative due to the ReLU non-linearity and NMF is considered as a more powerful MD model than VQ. VQ is known more as a data compression algorithm than its MD's formulation. The MU rule for solving NMF is also a practice-tested algorithm in the past years.
>
> The third point is why we need to soften VQ and CD. It is because we need to make the MD model differentiable to back-propagate the gradient. A lower temperature $T$ means that the relaxed problem is closer to the original one. Note that it will not receive more gains but increase the variance during training if we further lower $T$.
>
> > The related work section is provided in the appendix.
> > Other works employing matrix decomposition in the context of deep learning, are not covered.
>
> **A3**: Thanks for the criticism. We move the related works section to the main text and add a discussion with other works employing the decomposition in the context of deep learning.
>
> > - Table 1, no details are provided for the metric used for the results.
> > - Table 6, it would be better to specify the difference between the two entries of HamGAN.
> > - The text in almost all figure is quite small.
>
> **A4**: We clarify the settings in Tab. 1 and distinguish the two in Tab. 6. Most figures are plotted again with larger texts.

---

### Official Review · AnonReviewer4 · 2020-10-28
**An honest work**

**Rating:** 8
**Confidence:** 4

**Review:**

This paper proposed a matrix decomposition-based method to capture spatial long range correlation in the neural network. The proposed method employs an optimization method, i.e. non-negative matric factorization, to reconstruct a low-rank embedding of the input data. The experimental results show that the proposed method outperforms various popular attention-based methods in recent years for various vision tasks, i.e. semantic segmentation and image generation. This paper is basically well written and easy to follow what they have done.

However, I have several concerns that I listed as follows.
- According to experimental results, I find the proposed matrix decomposition based method outperforms several attention based methods in mIoU for semantic segmentation and FIN for image generation. Nevertheless, the Parameters, FLOPs, memory and running time are only compared with Dual attention network. Please compare more attention based methods to verify the proposed method is more efficient attention based methods.
- The proposed method is similar with EMA-Net, especially for employing concept decomposition as the optimization algorithm. Please discuss the relation and difference between the proposed method and EMA-Net.
- I am very interested in the initialization of matrix decomposition. In the supplementary, authors only discuss the initialization of D, so what is the best initialization of C for non-negative matrix factorization? Besides, what is the warm start with online update?
- The figures 3 and 4 are low quality, the text of coordinate is too small.

After rebuttal:
I appreciate the authors' detailed response to my questions, which largely addresses my previous concerns.
It's very pleasure to reviewing this interesting, innovative and well-written paper.
A clear accept.

---

> ### Author Response · Authors · 2020-11-24
> **Response to R4 (1/2)**
>
> We sincerely thank Reviewer 4 for perusing this work and understanding its technical details. Also, it is an honor to be commented as *An Honest Work*.
>
> > Please compare more attention based methods to verify the proposed method is more efficient attention based methods.
>
> **A1**: We will provide comparisons with more attention based methods in the revised version. Note that there is a trade-off between accuracy and cost. Hamburger can further lower its number of parameters and MACs via smaller $d$ and $r$, and can also get accelerated by cuttings down the number of iterations $K$, sacrificing little performance as the results already shown in the ablation experiments.
>
> > Please discuss the relation and difference between the proposed method and EMA-Net.
>
> **A2**: In the following paragraphs, we first discuss the relation between the two, then point out the differences. We add a new paragraph to the related works in the revised version investigating the low-rank exploration of attention-related global context modules.
>
>    1. The similarity between the two comes from the reformulation of the standard self-attention in the EMANet [1], in which attention-like operation is understood as an E-step on a mixture of vMF distribution [2]. Here the critical point is that the soft spherical K-Means algorithm in CD is equivalent to the EM inference on the mixture of vMF distribution, leading to the details found by R4. In fact, EMA Unit lies between VQ and CD, but it is not an MD model, because it employs a dictionary with unit atoms like CD but tries to reconstruct the representations in a VQ manner, losing the scale information hidden in the representations. Imagine replacing each hyper-pixel in the representations whose $L_2$ norm is not 1 but about 20 (observations from the experiments in the ablation settings) by the unit vector in the dictionary, which is what EMA Unit did.
>
>    2. The differences between this work and EMANet are twofolds, including both the framework and techniques.
>       - The difference in the framework is substantial. EMANet tries to find a small set of bases to be attended, as previous works do, such as $A^2$ Net [3]. They use an online EM algorithm on a mixture of vMF distribution to construct such a set, i.e., EM attention uses an online moving average and normalization. Though the output tensor is low-rank (Double Attention from $A^2$ Net is also low-rank), EMANet does not clearly define such a prior as an optimization problem and thus lacks high-level guidance, so that when attending to these unit vectors, a VQ-style operation is employed in the EMA Unit, losing the scale information in the representations. Additionally, the EMA Unit suffers from the unstable gradient issue. Although EMANet is a clean, clear, and excellent work with insightful intuition, pushing the low-rank exploration in computer vision along $A^2$ Net to multi-step EM manner, **it lacks a unified framework to understand why low-rank property helps and how to model and impose such property into the architecture**. In contrast, this paper demonstrates two important factors in encoding the global context, i.e., the low-rank completion formulation for modeling the global context and the optimization-driven designing approach, which can help us understand why low-rankness takes effect.
>
>       - For the technical part, we show that NMF is a simple yet strong MD model for capturing the global context and propose the one-step gradient to facilitate the training of Hamburger, while EMA Unit may suffer from the unstable gradient backward through the EM algorithm. Reflected by performance statistics, Hamburger with NMF obtains 78.3(77.8) while EMA Unit with BPTT (as the original paper) reaches 77.0(76.6) and EMA Unit with the one-step gradient can achieve 77.5(76.8), in the ablation settings. Note that the no_grad trick adopted in the code base of EMANet can attain 77.2(76.7), still lower than the one-step grad. The EMA Unit benefits less than Hamburger from the one-step gradient because it takes fewer iterations.

---

> > ### Author Response · Authors · 2020-11-24
> > **Response to R4 (2/2)**
> >
> > > What is the best initialization of C for non-negative matrix factorization? Besides, what is the warm start with online update?
> >
> > **A3**: It is an interesting question since the initialization, somehow, can determine the converged factorization results because the cost function of NMF is non-convex. We use $\boldsymbol{C} = softmax(\frac{1}{T}cosine(\boldsymbol{D}, \boldsymbol{X}))$ to initialize $\boldsymbol{C}$ after initializing $\boldsymbol{D}$ from $U(0, 1)$ as K-means is usually applied for initializing NMF and this initialization for $\boldsymbol{C}$ is equivalent to one step update via Spherical K-means. A special reminder is that it is not suitable to initialize either $\boldsymbol{D}$ or $\boldsymbol{C}$ to values too close to 0 due to the property of *Multiplicative Update*. So the temperature $T$ is recommended to be a higher value like 1 in this initialization.
> >
> > Before R4 mentioned this problem, we have not explored more initialization methods for $\boldsymbol{C}$ in NMF. Hence we test several possible initialization methods for $\boldsymbol{C}$, including $C_{ij} \sim U(0, 1)$ with scores 77.8(77.6), $C_{ij} = \frac{1}{r}$ with scores 77.6(77.2). It is surprising that random initialization also works for $\boldsymbol{C}$.
> >
> > Also, Hamburger with NMF does not need the moving average in the EMA Unit, which is thorny to implement, especially for GANs on TPU. This approach, named as *warm start with an online update* in this paper, is useful for the dictionary with unit vectors like CD. In general, however, random initialization is good enough for all three selected MD models. A possible reason is that it can enforce the network to adapt to the results solved by different initializations during the training process, acting like an *inner augmentation*.
> >
> > > The figures 3 and 4 are low quality, the text of coordinate is too small.
> >
> > **A4**: We remake all the important figures in the paper, especially enlarging the text of the coordinate. Many Thanks!
> >
> > [1] Li, Xia et al. “Expectation-Maximization Attention Networks for Semantic Segmentation.” 2019 IEEE/CVF International Conference on Computer Vision (ICCV) (2019): 9166-9175.
> >
> > [2] Banerjee, A. et al. “Clustering on the Unit Hypersphere using von Mises-Fisher Distributions.” J. Mach. Learn. Res. 6 (2005): 1345-1382.
> >
> > [3] Chen, Y. et al. “A2-Nets: Double Attention Networks.” ArXiv abs/1810.11579 (2018)

---

### Author Response · Authors · 2020-11-24
**General Response**

Dear Reviewers and AC,

Thanks for evaluating this work from different perspectives. We will update the revised version based on the valuable suggestions. In this general response, we highlight the main changes:

- Following R3 and R4's suggestions, we plot important figures with larger texts.

- Following R1, R3, and R4's suggestions, we move the related works section to the main text, add a discussion about efficient attention modules in computer vision and natural language processing, and review ideas in combining matrix decomposition and deep learning.

- Following R2's suggestion, we provide stronger experiments, pushing the PASCAL VOC test score to 85.9 mIoU and updating FID on ImageNet to 14.77.

- Following R4's suggestion, we add a more detailed comparison with different attention-related context modules regarding the cost.

- Following R1 and R3's suggestions, we rephrase the statement about the proposed approach to make it more appropriate.

Thanks for the comprehensive reviews to help us refine this paper. Please let us know if there are any additional questions.

Best,

Anonymous Authors

---

### Decision · Program_Chairs · 2021-01-07
**Final Decision**

**Decision:**

Accept (Poster)

**Comment:**

This paper introduces an alternative to self-attention, based on matrix factorization, and apply it to computer vision problems such as semantic segmentation. The method is simple and novel and obtains competitive results compared to existing approaches. The reviewers found the paper well written and easy to understand. For these reasons, I recommend to accept the paper.